# Effects of Different Silicon Sources on the Properties of Geopolymer Planting Concrete Mixed with Red Mud

**Wei Chen and Jiusu Li \***

School of Traffic and Transportation Engineering, Changsha University of Science & Technology, Changsha 410114, China

\* Correspondence: lijiusu@126.com

**Abstract:** In order to reduce the carbon emission of planting concrete in the process of preparation, and to realize the application of large amounts of red mud in the geopolymer, in this study, high silicon source materials and blast furnace slag are added to a large content of red mud base geopolymer planting concrete, which can remove the dependence of planting concrete on ordinary Portland cement and provide a new direction for the comprehensive utilization of red mud. In the paper, the effects of different A (Alkali solid content)/P (Powder dosage) and high silica sources (silica fume and diatomite) on the microstructure and fluidity of the geopolymer, as well as the compressive strength, pore characteristics, and alkalinity of the planting concrete, are comparatively evaluated. The corresponding results showed that when A/P was 0.25, the planting performance of the planting concrete would be reduced due to its high alkalinity; when A/P was 0.15, the planting concrete would have its sedimentation and the compressive strength decreased. On the other hand, the pozzolanic reaction among the silica fume, diatomite, and $Ca(OH)_2$ significantly weakened the alkali pan phenomenon in the later stage of planting concrete formation. The addition of an appropriate amount of silica fume and diatomite also made the structure of the geopolymer more compact with better fluidity, which yielded superior pore characteristics and planting performance for the planting concrete. For good planting concrete pore characteristics, the test results showed that the fluidity of the mortar should be 112–128 mm. Overall, the best planting concrete performance was achieved at an A/P ratio of 0.2, with the contents of silica fume and diatomite being 10% and 5%, respectively. Furthermore, the slope finite element analysis showed that planting concrete made with red mud geopolymer had better slope protection potential than ordinary Portland cement.

**Keywords:** red mud; silica fume (SF); diatomite (DA); geopolymer; planting concrete (PC)

## 1. Introduction

As an environment-friendly material, planting concrete (PC) can be used for slope protection to avoid the direct damage of the load to the subsoil of PC, but to also reduce the loss rate of the subsoil of the PC and increase the shear resistance of the soil [1]. Traditional planting concrete (PC) uses ordinary Portland cement (OPC) as the cementitious material. Although it has the inherent advantages of high strength and good stability, it is fairly resource intensive and often releases multiple greenhouse gases during the production process [2–4]. From the literature, it is reported that each ton of OPC will consume about 1.2 tons of limestone, 0.11 tons of standard coal, and emit about 0.85–0.92 tons of $CO_2$ and a significant quantity of $NO_x$ [5–7].

Compared with OPC, the raw materials for geopolymer usually comprise industrial waste without high-temperature calcination or sintering, which can reduce carbon emissions by about 70% during production and use [8]. Its high strength (compressive strength up to 15–74 MPa), high-temperature resistance (over 1000 °C), acid and alkali salt corrosion resistance (stable in alkaline solution and brine, acid resistance is better than OPC), and low permeability (close to OPC) properties make it an excellent OPC substitute [9–11].

With the rapid development of China's infrastructure field in recent years, industrial wastes such as metakaolin, fly ash, and blast furnace slag (BFS) have gained widespread usage in the preparation of cement, geopolymer, and other cementitious materials, with a corresponding price increase year by year [2,3]. As an industrial by-product produced when extracting alumina, red mud is strongly alkaline and contains heavy metals that will seriously pollute the surrounding water resources under the scouring of rain, while in a dry environment, its dust will escape to the atmosphere with the wind, endangering human health, so it belongs to harmful waste residues. By 2022, although China's accumulated stack of red mud has reached 110 million tons, its utilization rate is less than 5%, with local stacking being the most common treatment method for the red mud [12,13]. If red mud can be used to prepare geopolymer, it can not only reduce the cost of geopolymer, but also reduce the harm of red mud to the environment [14].

Red mud has high $SiO_2$ and $Al_2O_3$ contents, which meets the conditions for preparing geopolymer [15]. Red mud could also use its own alkalinity to reduce the amount of the alkali activator and reduce the alkalinity of concrete [16–18]. However, from the 24 h precipitation test of red mud soaked in NaOH solution, it could be seen that the maximum dissolution rates of aluminum and silicon in red mud were only 4% and 17%, respectively—far lower than the 40% and 50% of metakaolin [19].

If red mud is used to prepare geopolymer, it is often challenging to form a three-dimensional aluminum oxygen silicate network using the standard oxygen alternating bonding $[SiO_4]^{4-}$ and $[AlO_4]^{5-}$ tetrahedron, with a resultant poor compressive strength, flexural strength, and weather resistance [20,21]. Therefore, BFS is usually added to increase the polymerization degree of the red mud geopolymer [22]. BFS is used to improve the precipitation of silicon, aluminum, and calcium in the matrix system [23]. Due to the low bond energy, Ca-O will first break and release $Ca^{2+}$. $Ca^{2+}$ is easily formed with -(C-A-S-H)-gel structure using free $[SiO_4]^{4-}$ and $[AlO_4]^{5-}$ [24,25]. The formation of calcium–water compounds can enhance the strength of the polymer in red mud. However, with an increase in the BFS content, on the other hand, excessively high $Ca^{2+}$ content will also lead to the production of water-soluble $Ca(OH)_2$. This can ultimately reduce the early alkalinity of the geopolymer, which is a later stage efflorescence effect that is not conducive for the growth of plants.

High silicon sources have been proved to improve the compactness of calcium-based geopolymer structures, leading to a reduction in dry shrinkage cracking of the geopolymer due to water loss [26,27].

With this background, the work presented in this paper studied and evaluated the improvement of geopolymer alkalinity and PC planting performance using high silicon sources. In the study, two high silicon sources, namely, silica fume (SF) and diatomite (DA) [28–31], respectively, were selected to replace BFS in different proportions to comparatively evaluate the effects of different high silicon sources on the properties of geopolymer-based PC. To better reduce the alkalinity of the geopolymer, this study used A (alkali solid content)/P (powder dosage) as one of the variables to quantify the performance variations of the geopolymer-based PC as a function of different A/P ratios. The design of the research objective is to reduce the dependence on OPC and realize the application of high content of red mud. By adding BFS to improve the strength of PC and taking advantage of the excellent pozzolanic effect and high specific surface area characteristics of high silicon source materials to overcome the defects of BFS, we achieve the use of a high content of red mud base geopolymer to prepare PC, which will provide a new direction for the comprehensive utilization of red mud.

In terms of the paper organization, the subsequent section discusses the materials, laboratory tests, and the analysis methods used in the study. Results are thereafter presented, analyzed, and discussed. The paper then concludes with a synthesis of the results, summary of key findings, and recommendations.

## 2. Experimental Matrix Plan

### 2.1. Materials Used

Bayer processed red mud, BFS, SF, and DA were heated in an oven at 80 °C for 1 day to remove water (all materials are from the Henan Province of China), and then sealed and stored at room temperature. The chemical composition is shown in Table 1. The morphology of the SF and DA materials is photographically shown in Figure 1.

**Table 1.** Chemical composition of the cementitious materials.

|  | **Red Mud (%)** | **BFS (%)** | **DA (%)** | **SF (%)** |
|---|---|---|---|---|
| $SiO_2$/% | 26.8 | 34.5 | 82.3 | 94.8 |
| $Al_2O_3$/% | 24.0 | 17.7 | 6.7 | 0.81 |
| CaO/% | 7.0 | 34.0 | 0.79 | 1.86 |
| $Fe_2O_3$/% | 26.8 | 1.03 | 1.98 | 0.08 |
| MgO/% | - | 6.01 | 0.34 | 0.65 |
| $Na_2O$/% | 11.8 | - | - | 0.45 |

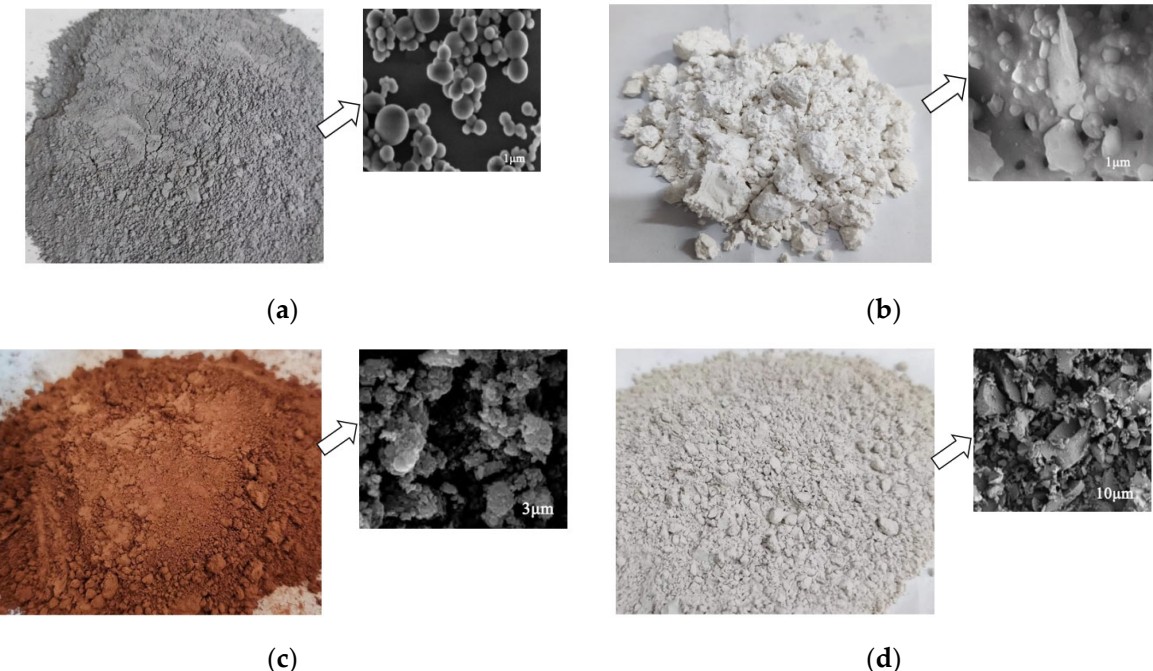

**Figure 1.** Morphology of and microstructure of the SF, DA, BFS, and red mud: (**a**) SF; (**b**) DA; (**c**) Red mud; (**d**) BFS.

The morphology and microstructure of the SF, DA, BFS, and red mud are photographically shown in Figure 1. It can be seen from Figure 1 that SF is a kind of spherical particle, and the dispersion is relatively uniform. DA has many pores, its structure is relatively compact, and there are many fine particles on the surface. Red mud and BFS are irregular in shape, loose in arrangement, and rough in surface, and red mud is reddish brown due to a large amount of $Fe_2O_3$ [32]. However, there are many small particles in BFS adhering to large particles, showing a flaky structure. On the other hand, the loss on ignition of red mud, BFS, SF, and DA are 9.81%, 0.9%, 2.8%, and 18% respectively. The specific surface areas are 64 $m^2$/g, 5 $m^2$/g, 24 $m^2$/g, and 18 $m^2$/g. The density is 2400 $kg/m^3$, 2900 $kg/m^3$, 2100 $kg/m^3$, 470 $kg/m^3$, respectively.

The alkali activator comprised of sodium silicate with a modulus ratio ($SiO_2/Na_2O$) of 3.3, analytical pure granular NaOH with a purity of 98%, and an appropriate amount of water. After preparation, it was sealed and stored for 1 day at room temperature (to ensure complete the dissolution of NaOH). The aggregate used was basalt with a gradation of

13.2–16 mm (stored after cleaning and drying). Humus and natural soils were used along with the addition of pine needles (ground through a 0.6 mm sieve) to increase the acidity of the soil and weaken the negative impact of alkaline substances in geopolymers on the soil. Ryegrass was selected and used as the plant type for the study.

### 2.2. Sample Preparation and Replicates

Prior to adding aggregates, the mixer was first moisturized. Thereafter, the powder (i.e., a powdery blend of BFS, red mud, SF, and DA) was added and mixed evenly. Finally, the alkali activator with modulus of 1.2 and water were poured with a water/powder ratio of 0.3. The PC mix proportions are shown in Table 2. The relevant preparation and test flow chart is shown in Figure 2.

**Table 2.** Mix proportions of PC.

| Sample No. | Red Mud (kg/m$^3$) | SF (kg/m$^3$) | DA (kg/m$^3$) | BFS (kg/m$^3$) | Aggregate (kg/m$^3$) | A/P | Water (kg/m$^3$) |
|---|---|---|---|---|---|---|---|
| A1 | 115 | 0 | 0 | 115 | 1500 | 0.15 | 45.75 |
| A2 | 115 | 11.5 | 0 | 103.5 | 1500 | 0.15 | 45.75 |
| A3 | 115 | 23 | 0 | 92 | 1500 | 0.15 | 45.75 |
| A4 | 115 | 34.5 | 0 | 80.5 | 1500 | 0.15 | 45.75 |
| A5 | 115 | 46 | 0 | 69 | 1500 | 0.15 | 45.75 |
| a1 | 115 | 0 | 11.5 | 103.5 | 1500 | 0.15 | 45.75 |
| a2 | 115 | 0 | 23 | 92 | 1500 | 0.15 | 45.75 |
| a3 | 115 | 0 | 34.5 | 80.5 | 1500 | 0.15 | 45.75 |
| a4 | 115 | 0 | 46 | 69 | 1500 | 0.15 | 45.75 |
| B1 | 115 | 0 | 0 | 115 | 1500 | 0.20 | 38.00 |
| B2 | 115 | 11.5 | 0 | 103.5 | 1500 | 0.20 | 38.00 |
| B3 | 115 | 23 | 0 | 92 | 1500 | 0.20 | 38.00 |
| B4 | 115 | 34.5 | 0 | 80.5 | 1500 | 0.20 | 38.00 |
| B5 | 115 | 46 | 0 | 69 | 1500 | 0.20 | 38.00 |
| b1 | 115 | 0 | 11.5 | 103.5 | 1500 | 0.20 | 38.00 |
| b2 | 115 | 0 | 23 | 92 | 1500 | 0.20 | 38.00 |
| b3 | 115 | 0 | 34.5 | 80.5 | 1500 | 0.20 | 38.00 |
| b4 | 115 | 0 | 46 | 69 | 1500 | 0.20 | 38.00 |
| C1 | 115 | 0 | 0 | 115 | 1500 | 0.25 | 30.25 |
| C2 | 115 | 11.5 | 0 | 103.5 | 1500 | 0.25 | 30.25 |
| C3 | 115 | 23 | 0 | 92 | 1500 | 0.25 | 30.25 |
| C4 | 115 | 34.5 | 0 | 80.5 | 1500 | 0.25 | 30.25 |
| C5 | 115 | 46 | 0 | 69 | 1500 | 0.25 | 30.25 |
| c1 | 115 | 0 | 11.5 | 103.5 | 1500 | 0.25 | 30.25 |
| c2 | 115 | 0 | 23 | 92 | 1500 | 0.25 | 30.25 |
| c3 | 115 | 0 | 34.5 | 80.5 | 1500 | 0.25 | 30.25 |
| c4 | 115 | 0 | 46 | 69 | 1500 | 0.25 | 30.25 |

Once the aggregate surfaces presented a glass luster and there was no agglomeration of the cementitious materials, the PC mix was poured into a 100 mm × 100 mm × 100 mm test mold in three layers using the inserting and tamping molding method [33–35]. The samples were then cured for 3, 7, and 28 days, respectively, at 80 °C and 98% humidity. Based on the above experimental procedures, numerous PC samples with different SF and DA contents at different A/P ratios were prepared. At minimum, three replicate samples were prepared per material type per A/P ratio per mix proportions.

### 2.3. Laboratory Testing Methods

2.3.1. Mineralogical Characteristics and Microstructure

A D8advance A25 X-ray (Bruker, USA) diffractometer was used to analyze the phase morphology of the red mud geopolymer prepared with different amounts of SF and DA after 7 days of curing. A Hitachi S-4800 field emission scanning electron microscope

(Hitachi, Japan) was used to observe the development of cracks and pores of red mud geopolymer cured for 7 days with different contents of SF and DA.

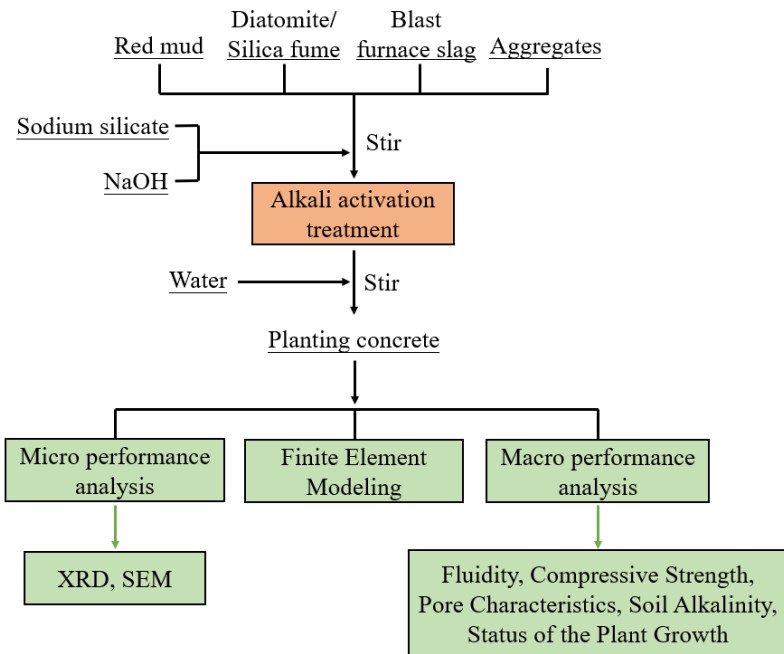

**Figure 2.** Flow chart of this study.

### 2.3.2. Fluidity

The fluidity of the geopolymer mortar with different SF and DA contents was determined by measuring the flow diameter of the geopolymer using a cone (60 mm high, 36 mm top diameter, and 60 mm bottom diameter). Test 3 test pieces for each mix proportion and take the middle value.

### 2.3.3. Compressive Strength

The compressive strength of the PC was determined based on the Chinese test method JTG E30-2005 [36]. After curing for 3 days, 7 days, and 28 days, the compressive strength test shall be carried out immediately after curing, and the pressure shall be controlled at 0.3–1.0 MPa/s. Because non-standard test pieces are used, the conversion factor of 0.95 should be multiplied when calculating the compressive strength of test pieces to offset the strength error caused by size [36]. The samples were cubically shaped with dimensions of $100 \times 100 \times 100$ mm. Three test pieces shall be tested for each mix proportion. When the difference between the strength value of the test pieces and the arithmetic mean value is less than 20%, the intermediate value shall be taken. Otherwise, the test shall be repeated.

### 2.3.4. Pore Characteristics

After curing for 7 days, the samples were put into an oven at 80 °C to dry to constant weight and we recorded the dry weight of the sample thereafter. Then, the water purification balance method was used to measure the mass of the samples in water [37]. The connected porosity of the sample was calculated using Equation (1). Three test pieces shall be tested for each mix proportion. When the difference between the maximum value and the minimum value and the median value is not more than 15% of the median value, the median value shall be taken. Otherwise, the test shall be repeated.

$$C_V = \left(1 - \frac{m_2 - m_1}{\rho v}\right) \times 100\% \tag{1}$$

where $C_V$ is the connected porosity of the sample (%); $m_1$ is the mass of the sample in water (g); $m_2$ is the mass of the sample after curing (g); $\rho$ is the density of water (g/cm$^3$); and v is the volume of the sample (cm$^3$).

Sand particles with an average particle size of 2.36~4.75 mm were selected using the sieve analysis method [37]. After cleaning and drying, 100 g sand particles were selected, evenly sprinkled on the samples, and then, turned on the vibrator. A square hole sieve with a hole diameter greater than 4.75 mm was placed at the bottom of samples. The machine was stopped after vibrating for 5 min. The planting porosity of the samples was calculated (the proportion of pores with appropriate hole diameter and connected pores) using Equation (2). Three test pieces shall be tested for each mix proportion. When the difference between the maximum value and the minimum value and the median value is not more than 15% of the median value, the median value shall be taken. Otherwise, the test shall be repeated.

$$C_d = \frac{m_s}{100} \times 100\% \qquad (2)$$

where $C_d$ is the planting porosity of the sample (%); and $m_s$ is the total mass of sand passing the square hole sieve (g).

### 2.3.5. Soil Alkalinity

Humus soil with pine needle powder was filled into the sample cured for 7 days; thereafter, it was allowed to cure at room temperature for 1 day, 7 days, and 28 days, respectively. Because the soil directly contacts the alkaline substances on the PC surface, the alkaline substances on the PC surface will also dissolve into the soil due to the migration of water. Therefore, this test determines whether the internal environment of PC is suitable for plant growth by testing the alkalinity of the soil. We took 10 g soil in the center of the sample and ground it into a 2 mm sieve. After drying, the sample matrix was put in a 50 mL beaker. About 25 mL of deionized water was added and stirred for 5 min. After stabilizing for 1 h, the alkalinity of the soil was measured. Three test pieces shall be tested for each mix proportion, and the arithmetic means value shall be taken.

### 2.3.6. Status of the Plant Growth

After the humus soil and water were prepared in a ratio of 1:0.8, the fluid matrix was poured into the sample that had been cured for 7 days until the liquid exuded from the bottom and covered the sample completely. After drying, 1 cm thick natural soil was paved onto the sample. We placed the test piece in the room of 20–25 °C for cultivation. Thereafter, grass seeds were planted, and 1 cm thick of natural soil was added again. The PC was broken on the 20th and 40th days, respectively. The plant rhizome length was measured to evaluate the growth status of the plant so as to verify the planting performance of PC, as exemplified in Figure 3. Three test pieces shall be tested for each mix proportion. We shall select 10 rhizomes at the center of each test piece, and the middle value shall be taken.

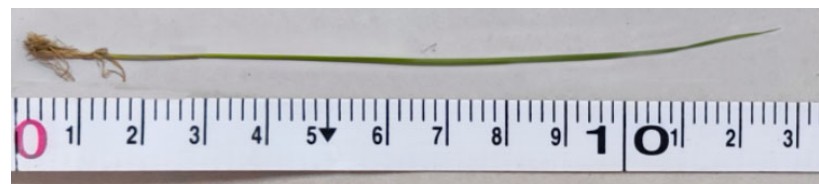

**Figure 3.** Measurement of the plant rhizome length.

### 2.3.7. Finite Element Modeling

ABAQUS 2020 finite element software was used to establish the model for finite element (FE) analysis. The slope stability with PC slope protection was analyzed. The slope gradient was assumed to be 60°, and the height was 15 m. Relevant parameters of the simulated soil for FE modeling are shown in Table 3.

**Table 3.** Soil material parameters for FE modeling.

| | Unit Weight (kn/m$^3$) | Saturated Volume Water Rate/% | Saturated Permeability Coefficient/(m/s) | Cohesion/kPa | Internal Friction Angle/(°) | Elastic Modulus/Kn/m$^3$ | Poisson's Ratio |
|---|---|---|---|---|---|---|---|
| Soil | 19.2 | 25 | $5.4 \times 10^{-6}$ | 24 | 32.2 | 4000 | 0.3 |
| PC | 20 | - | - | - | - | 16,000 | 0.2 |

## 3. Results, Analysis, and Discussions

### 3.1. Mineralogical Characteristics and Microstructure

3.1.1. Mineralogical Characteristics Analysis

Figure 4 shows the XRD patterns of geopolymers with different SF and DA contents. At 25°–35° 2θ, there are two very sharp oblique Calcium boiling stone peaks and boiling stone peaks. There is an obvious Ca(OH)$_2$ amorphous hump at 42.5° 2θ. With an increase in the SF and DA content, the Calcium content decreased, resulting in a gradual enhancement of the zeolite diffraction peak, a decrease in the wairakite peak intensity, and a flattening of Ca(OH)$_2$ amorphous hump. This indicates that the zeolite phase replaces part of the wairakite phase, and the free Ca$^{2+}$ is also relatively reduced.

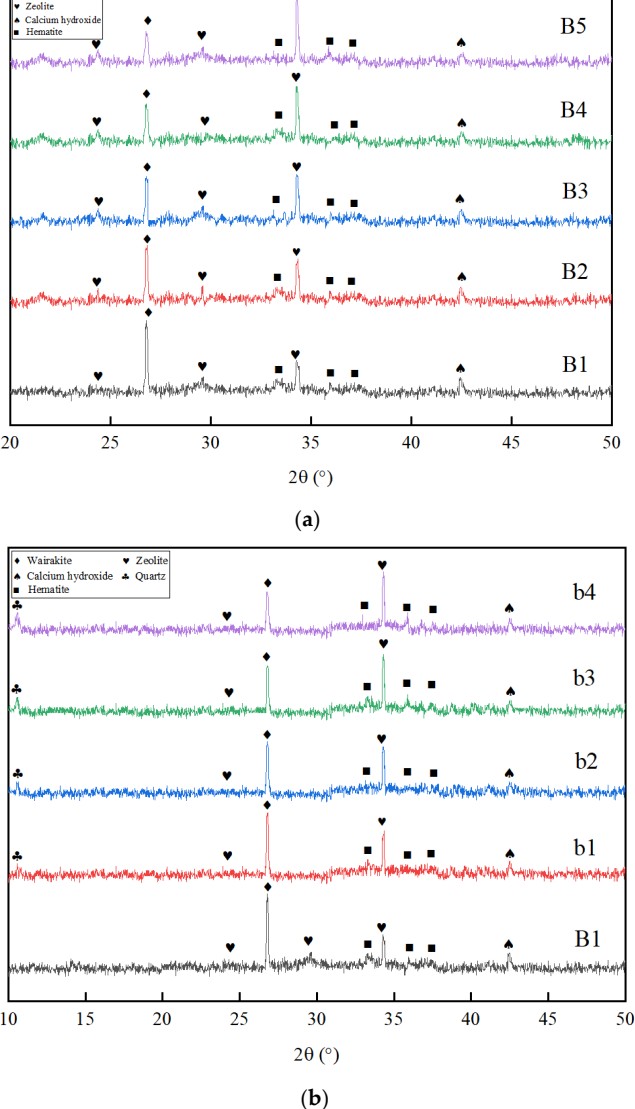

**Figure 4.** XRD patterns of geopolymers prepared from different silicon sources: (**a**) SF; (**b**) DA.

Figure 4b also shows that compared with SF, the addition of DA weakens the intensity degree of the wairakite and zeolite peaks. The incorporation of DA will produce a tiny amorphous hump of the quartz phase at 10°–15° 2θ, and the quartz diffraction peak becomes more and more evident with an increase in the DA content, indicating that more DA was not involved in the chemical reaction. On the other hand, in Figure 4a,b, it is shown that hematite exists at 30°–40° 2θ, and its peak value changes little, indicating that it may not participate in the geopolymer reaction. Relevant research also proves this point [32].

### 3.1.2. Microstructure Analysis

Figure 5 shows the microstructure changes of geopolymers without high silicon source materials at different A/P ratios. When the A/P ratio is low, the chemical reaction rate is also slow. Under this condition, the overall structure of the geopolymer is loose and many fine pores will be produced. By contrast, the polymerization rate accelerates with an increase in the A/P ratio, resulting in the water loss rate of the geopolymer being accelerated and many dry shrinkage cracks appearing.

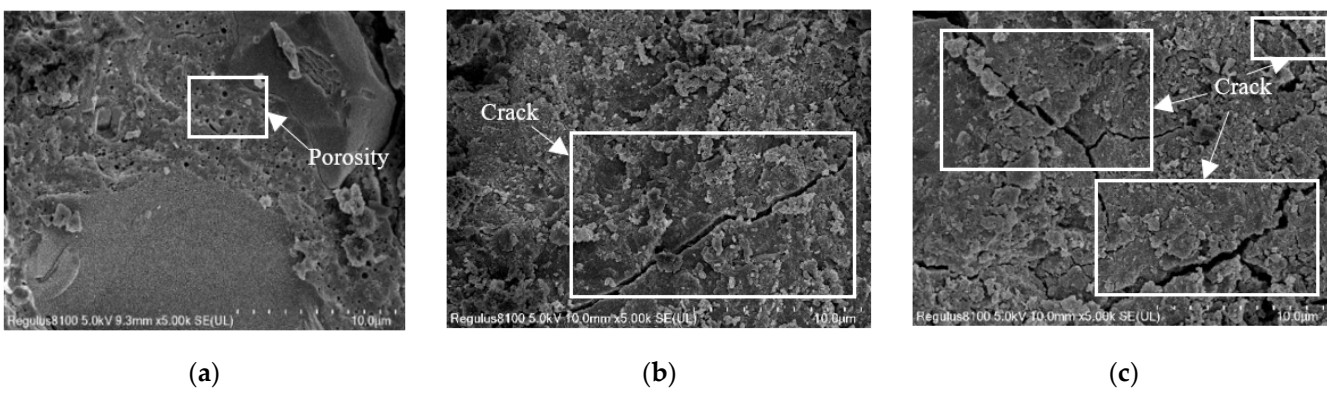

(**a**)  (**b**)  (**c**)

**Figure 5.** SEM images of geopolymer at different A/P ratios: (**a**) 0.15; (**b**) 0.20; (**c**) 0.25.

SF and DA have large specific surface areas to fill the tiny pores in the structure to reduce the dry shrinkage cracks caused by pore water loss. As can be seen from Figure 6a,b, with the increase of silica fume content, the cracks gradually narrow. Adding 5% diatomite has a similar effect. However, it can be seen from Figure 6c,f that when the content of diatomite increases to 10%, the dry shrinkage cracks in the structure will expand, and a large number of circular aggregates will appear with the increase of the content. Considering the high-water absorption capacity and plasticity of DA, when the DA content is too large, it is difficult to disperse evenly and will accumulate in the slurry. With an increase in the DA content, the number of aggregates gradually decreased whilst the volume progressively increased.

### 3.2. Fluidity Analysis of the Geopolymer Slurry

Figure 7 shows that increasing the A/P ratio will reduce the fluidity of the geopolymer slurry. The greater the A/P ratio, the more intense the polymerization, which inherently accelerates the generation of the gel structure, resulting in a decrease in fluidity.

At the same A/P ratio, SF can significantly improve the fluidity of the slurry, with the greatest gain occurring at 10% SF (the maximum increase of 19%). It is apparent that the low content of SF can have a significant morphological effect, including acting as balls in the slurry, filling the micropores, releasing the water in the pores, and ultimately improving the fluidity of the slurry. However, the specific surface area of SF is considerably large, and the excessive amount of SF will exacerbate the overall water absorption of the material matrix and reduce the fluidity of the slurry. The specific surface area of DA is much larger than that of SF, and only 5% DA can produce an impactful morphological effect. However, more alkali-soluble silicon in the DA can significantly improve the polymerization reaction of the gelling system. Additionally, its high-water absorption capacity and high plasticity

also make it easy to accumulate in the slurry, resulting in a significant reduction in the overall fluidity of the slurry.

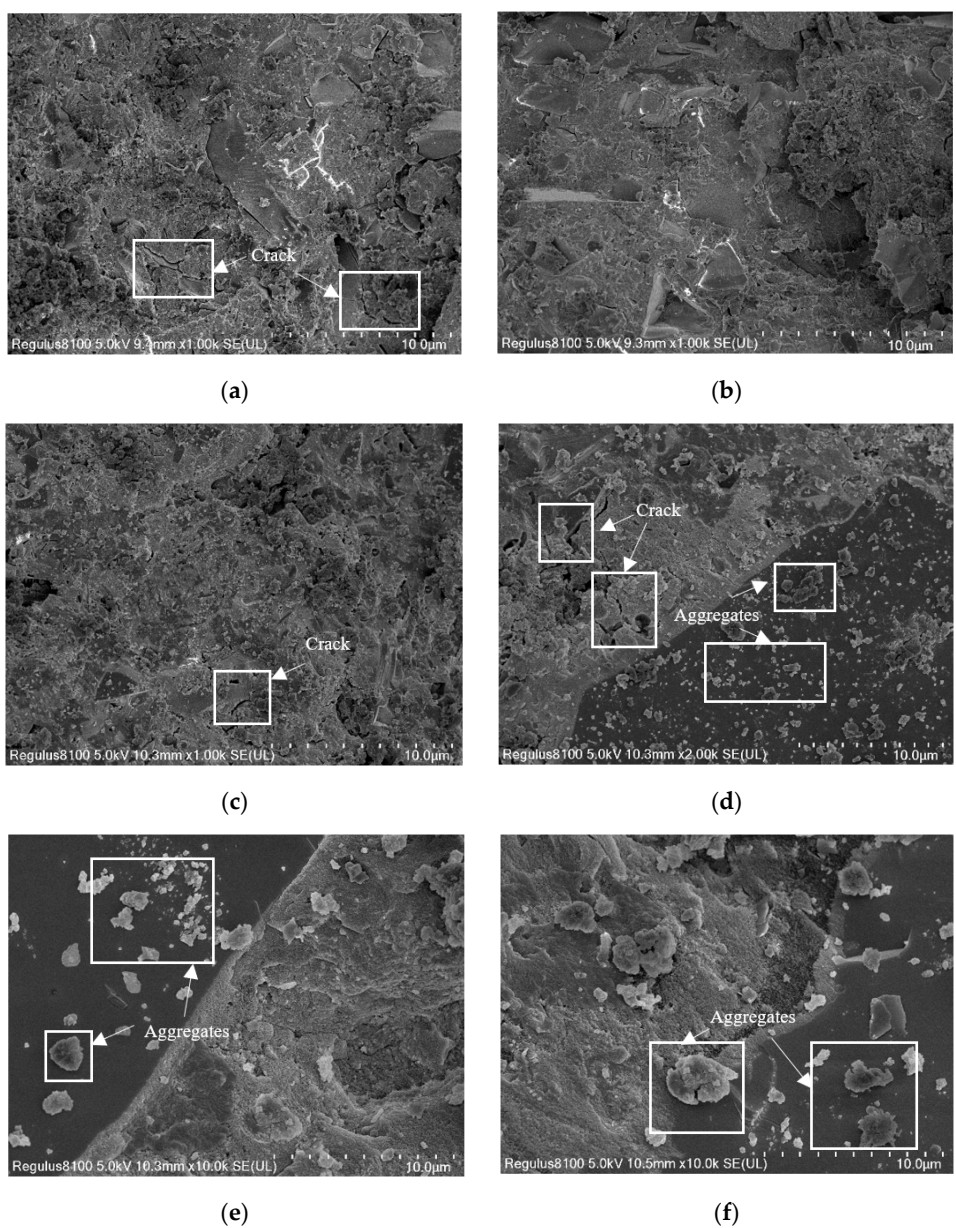

**Figure 6.** SEM images at different SF and DA contents for A/P = 0.2: (**a**) 5% SF; (**b**) 10% SF; (**c**) 5% DA; (**d**) 10% DA; (**e**) 15% DA; (**f**) 20% DA.

### 3.3. Analysis of the PC Mechanical Properties

In general, an increase in the A/P ratio will accelerate the dissolution rate of silicon and aluminum and promote the formation of -(N-A-S-H)- and -(C-A-S-H)- gel. This is exemplified by the change in the compressive strength in Figure 8. However, when the A/P ratios increases from 0.2 to 0.25, the gain in the PC compressive strength decreases from about 50% to less than 10%. Because more $OH^-$ will form water-soluble $Ca(OH)_2$ with the dissolved $Ca^{2+}$ in the matrix system, the formation of the gel will lead to the loosening of the structure and accelerate the rate of water loss, thus increasing the dry shrinkage deformation of the polymer [38] that will, in turn, negate the evolution of the PC compressive strength. With the addition of SF and DA, however, the effect is alleviated, but not significantly noticeable.

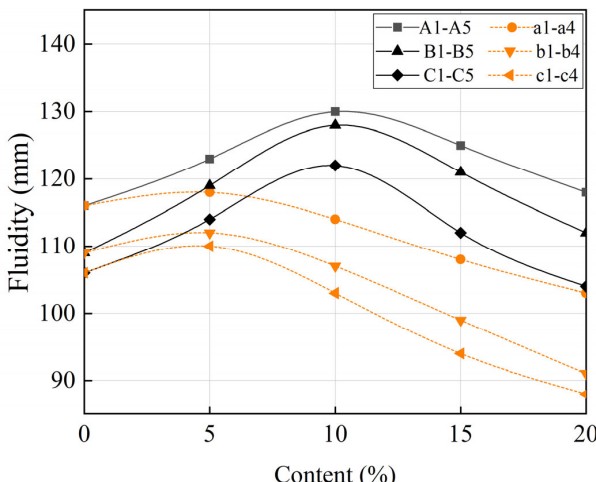

**Figure 7.** Effects of different SF and DA contents on the fluidity of slurry at different A/P ratios.

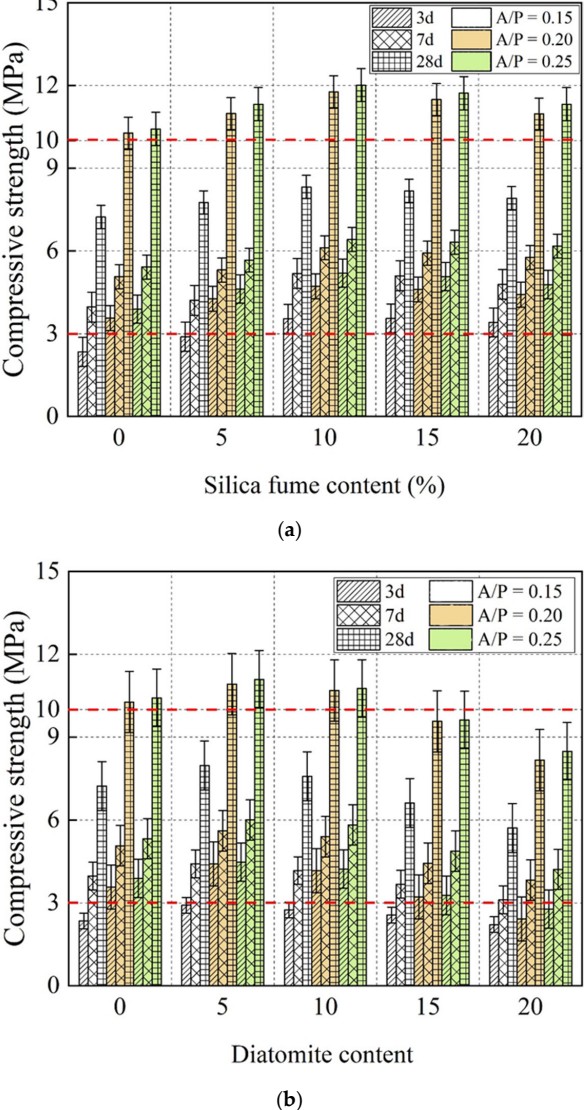

**Figure 8.** Compressive strength of PC with different contents of SF and DA contents at different A/P ratios: (**a**) SF; (**b**) DA (above the red line is better than the standard of JC/T 2557-2020 [37]. The 7-day compressive strength is greater than 3 MPa and the 28-day compressive strength is greater than 10 MPa).

In Figure 8, the compressive strength of the PC increases first and then decreases with the change in the SF and DA content, reaching maximum values at 10% and 5%, respectively, with day 7 to day 28 being the greatest increment in the PC compressive strength. The main reason is that adding SF and DA will increase the precipitation of silicon and aluminum elements in the matrix system. The large specific surface area of SF and DA will also inherently accelerate the polymerization of the matrix system and reduce the formation of water-soluble $Ca(OH)_2$. SF and DA can also rely on their pozzolanic effects to react with $Ca(OH)_2$ to produce hydrated calcium silicate [39,40], thus making the structure more compact. However, the metal activity of SF and DA is relatively low [40,41], so the early polymerization reaction mainly relies on BFS. In the later stage of the chemical reaction, SF and DA will provide the silicon and aluminum elements that are required for the reaction, thus significantly improving the compressive strength of the PC. This has also been confirmed in previous studies [26].

Compared with the -(N-A-S-H)- structure, the formation of -(C-A-S-H)- can make the structure of cementitious material more compact [42]. However, when the substitution amount of SF for BFS exceeds 10%, the PC compressive strength decreases only slightly. Considering that silicon replaces the calcium in the cementitious matrix system, the total amount of calcium is reduced by less than 7%. However, the total amount of silicon elements was observed to have increased by nearly 19%. Therefore, when the content of SF is 10–20%, the effect on the PC compressive strength was marginal.

Although the difference in the elemental content is small, the high-water absorption capacity and high plasticity of DA makes it easy to absorb water and accumulate it in the slurry. So, it is easy to appear as stress concentrations and form weak points after PC molding. On the other hand, the sharp decline in the mortar fluidity makes the PC workability to worsen, i.e., the aggregate becomes difficult to be evenly wrapped by the mortar, and the adhesion between the aggregate particles declines, resulting in a significant compressive strength reduction.

### 3.4. Analysis of the PC Pore Characteristics

#### 3.4.1. Connected Porosity

Only the connected pores are suitable for plant growth. Otherwise, the plant roots will be blocked in the pores. When watering, caution should generally be excised as the water can easily accumulate above the needed volume, resulting in root hypoxia and possibly plant death. The connected porosity is related to the fluidity of the slurry. Figure 9 shows the influence of different slurry fluidity on PC. In general, the greater the slurry fluidity is, the more pronounced the slurry settling phenomenon is.

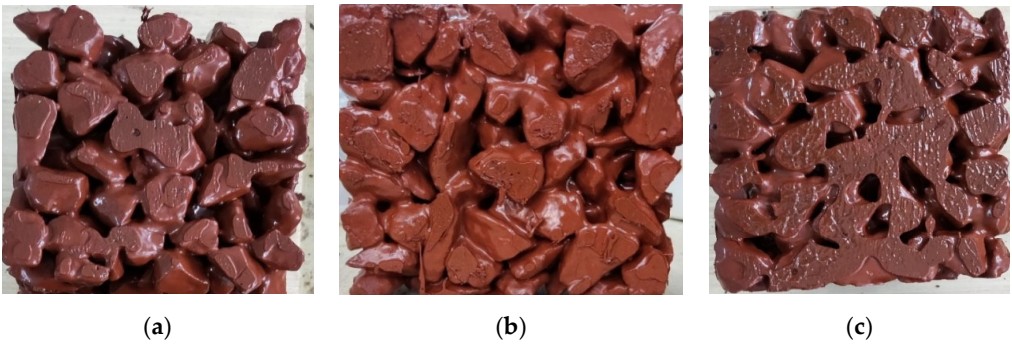

|     |     |     |
| :-: | :-: | :-: |
| (**a**) | (**b**) | (**c**) |

**Figure 9.** Sedimentation diagram of PC samples under different slurry fluidity (**a**) when the fluidity of the slurry is 105 mm; (**b**) when the fluidity of the slurry is 115 mm; (**c**) when the fluidity of the slurry is 130 mm.

By analyzing Figure 10 in combination with Figure 7, it can be seen that the increase in slurry fluidity will reduce the PC connectivity porosity. The PC mixed with SF clearly demonstrates this phenomenon. For the PC mixed with DA, when the DA content exceeds

10%, the high-water absorption of DA significantly reduces the slurry fluidity. Although there is no slurry settlement, a large amount of powder accumulation causes some of the pores in the PC to be blocked, leading to a reduction in the PC's connected porosity.

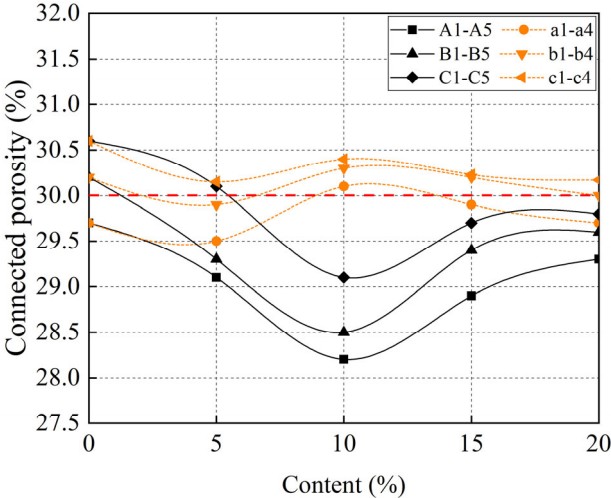

**Figure 10.** Connected porosity of PC with different contents of SF and DA under different A/P ratios (below the red line is better than the standard of JC/T 2557-2020 [37]).

On the premise of meeting the PC workability and considering the continuous decline in the slurry fluidity, the slurry that is adhered to the aggregate particles becomes thinner and thinner, with the PC connectivity porosity flattening out.

The correlation between the slurry's fluidity and the PC connection's porosity is fitted in Figure 11. The governing equation is:

$$y = -3.51x^3 + 0.01x^2 - 0.89x + 57.43. \tag{3}$$

After evaluating all data, $R^2$ can be regarded as 0.93, which means that there is a very strong relationship between the fluidity of the slurry and the connected porosity of the PC. With the decrease of the fluidity of the slurry, the connected porosity of the PC gradually tends to be flat. Therefore, the connected porosity of PC can be predicted by using the slurry fluidity according to Equation (3).

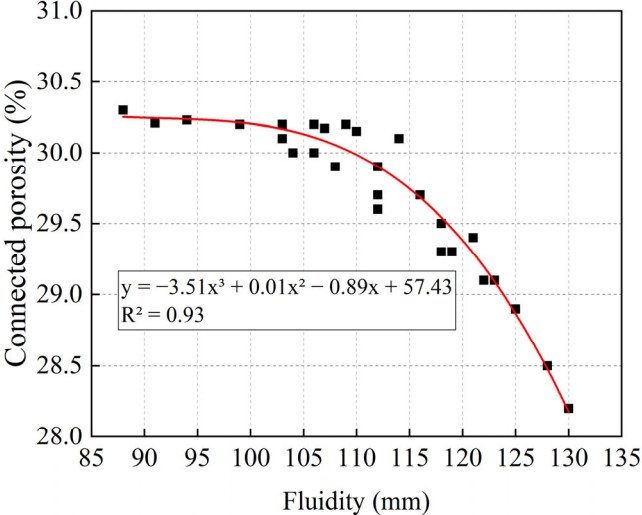

**Figure 11.** Correlation between the slurry fluidity and PC connected porosity.

### 3.4.2. Planting Porosity

In addition to requiring pore connectivity, the size of the pore diameter is more critical for the planting pores. Only connected pores with a pore diameter greater than 2.36 mm can ensure the expected plant growth. Figure 12 shows that the planting porosity of PC is positively correlated with the connectivity porosity; that is, the planting porosity increases with the increase of the connected porosity.

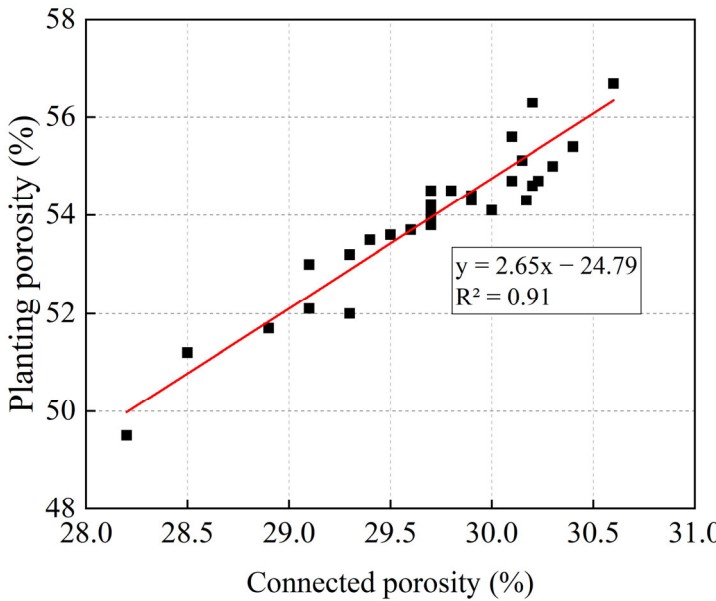

**Figure 12.** Correlation between the connected porosity and planting porosity of PC.

According to Figures 7 and 13, when the fluidity of the slurry is less than 128 mm, the planting porosity required by the specification (i.e., planting porosity greater than 50% [37]) can be met. With the change of SF content, the planting porosity first decreased and then increased. When the content of SF is not more than 10%, and the content of DA is not more than 5%, it can significantly improve the fluidity of the mortar. Under these conditions, the PC sedimentation will reduce the planting porosity. With an increase in the content of SF and DA, the fluidity of the slurry decreases, the slurry settling phenomenon disappears, and the planting porosity of PC increases. However, due to the high-water absorption of DA and SF, the consistency of the slurry gradually improves, with the slurry wrapped around the aggregate particles becoming thicker. The pore size of PC also becomes smaller so that an increase in the plant porosity of the PC becomes placid.

Especially for DA, too much DA will significantly reduce the fluidity of the slurry and make the slurry to accumulate inside the PC even if there is no slurry settlement, with the planting porosity exhibiting a downward trend (see Figures 7 and 13). This is partly attributed to the narrow internal pores. With the continuous decrease in the fluidity of the mortar, the downward trend tends to be gentle.

The relationship between the fluidity of the slurry and the porosity of PC is shown in Figure 14. $R^2$ is 0.86, indicating that there is a strong correlation between the fluidity of the slurry and the planting porosity of the PC. When the fluidity of the slurry decreases to a certain extent, the change of the planting porosity of the PC gradually tends to be flat. Therefore, according to Equation (4), it is possible to predict the planting porosity of PC by using the fluidity of the slurry.

$$y = -1.21e^{-4}x^3 + 0.03x^2 - 3.11x + 151.07 \tag{4}$$

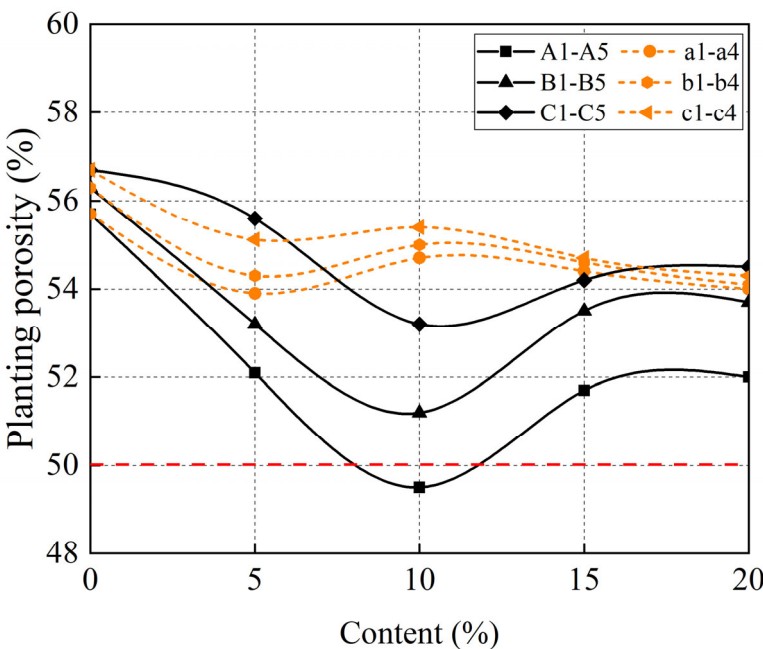

**Figure 13.** Planting porosity of PC with different SF and DA contents at different A/P ratios (above the red line is better than the standard of JC/T 2557-2020 [37]).

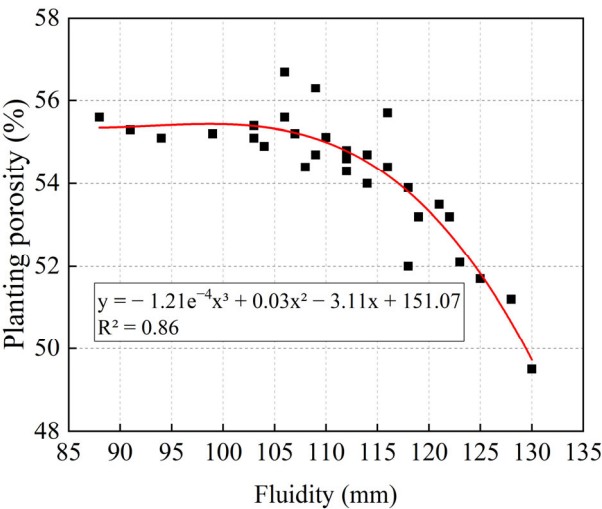

**Figure 14.** Correlation between slurry fluidity and planting porosity of PC.

### 3.5. Analysis of the Soil Alkalinity

Most plants cannot grow in soils with high alkalinity. For PC, the control of the soil alkalinity in pores is the key to the final survival of the plants. Pine needle humus soil is acidic, with a pH value of about 3–5 [43], which is conducive for weakening the impact of alkali diffusion in PC on the soil alkalinity.

The diffusion of alkali in PC takes time. With the extension of curing time, the residual $OH^-$ and water-soluble $Ca(OH)_2$ on the surface of the PC will gradually diffuse in the soil with water, thus increasing the soil alkalinity. When alkaline substances infiltrate the soil, the soil alkalinity is irreversible without the interference of external factors. Therefore, the alkalinity should be well controlled in the short term. Figure 15 confirms that the alkalinity of the test sample can meet the specification standard [37] only when the alkalinity difference of the test sample in the first 7 days is marginal. With an increase in the A/P ratio, the content of NaOH increases, and the alkalinity also increases significantly, particularly when the A/P ratio increases from 0.2 to 0.25.

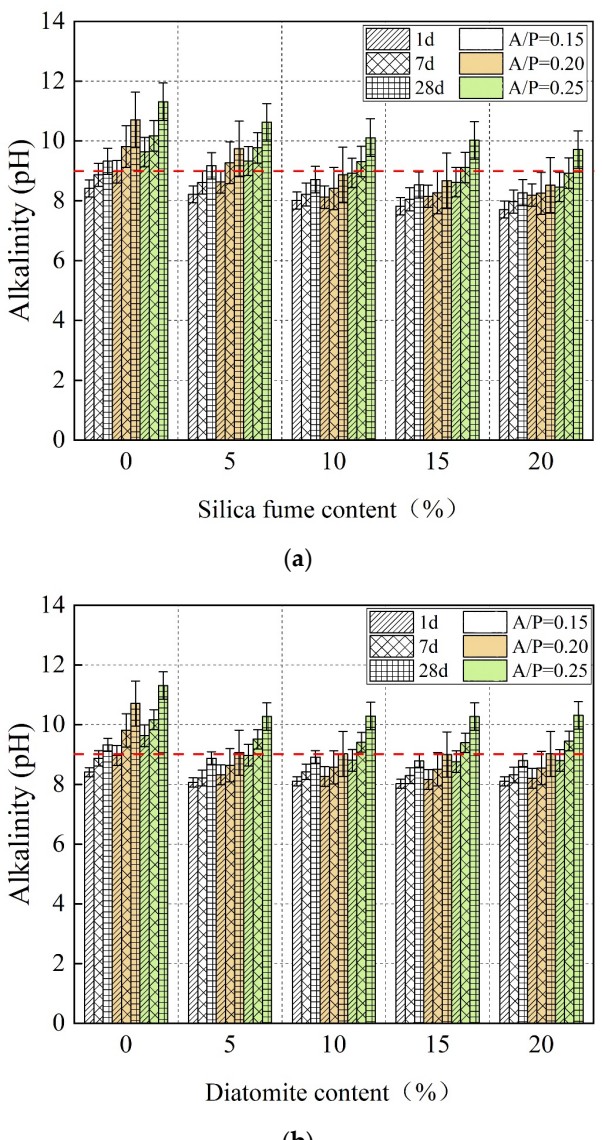

**Figure 15.** Alkalinity of soil inside the PC with different SF and DA contents at different A/P ratios: (**a**) SF; (**b**) DA (below the red line is better than the standard of JC/T 2557-2020 [37]).

Figure 15 shows that the addition of SF and DA will reduce the alkalinity of the soil. As previously mentioned, SF and DA can rely on their pozzolanic effects to consume the $Ca(OH)_2$ in the system [39,40], reducing the pan alkali phenomenon in the later stages of PC formation. It has been confirmed in the relevant research of PC that water-soluble $Ca(OH)_2$ is one of the reasons for the high alkalinity of PC [44,45]. However, Figure 14b shows that when the DA content is more than 10%, the soil alkalinity slightly increases on day 7 and day 28. This phenomenon is partially attributed to the decline in the accumulation of diatomite, which leads to the small diameter of some pores as well as high alkalinity in the pores.

### 3.6. Analysis of the PC Planting Performance

From the previously presented and discussed results, it was observed the PC's best performance occurred when the A/P ration is 0.2. Therefore, the planting performance of the PC was correspondingly evaluated at an A/P ratio of 0.2. Figure 16 shows that using geopolymer to prepare PC is a practical method that can reduce carbon emissions on the premise of ensuring the healthy plant growth.

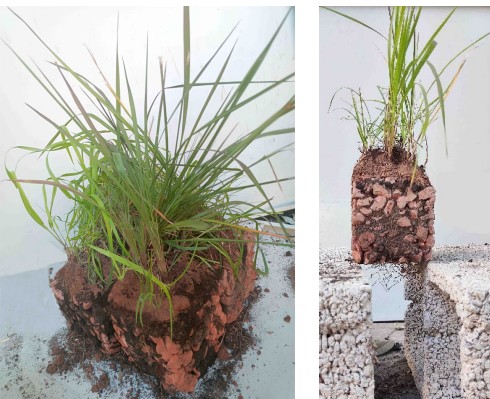

**Figure 16.** Plant status on the 40th day.

Figure 17 shows that due to the high alkalinity in the later stages of growth, plants in PC without SF and DA died in the next 20 days. However, an appropriate amount of SF and DA can improve the planting performance of PC. Provided that the alkalinity meets the requirements (i.e., the pH value of soil less than 9 in 28 days), the later plant growth mainly depends on the pore condition of PC. With a decrease in the connectivity porosity and plant porosity, the growth rate of plants slowed down. Figure 17a shows that the planting performance of PC increases with an increase in the SF content. When the SF content is more than 10%, the plant growth rate slows down because the change in the soil alkalinity is not apparent. Although the addition of SF will lead to the sedimentation of PC, it is not severe, and PC has enough connected pores to ensure plant growth.

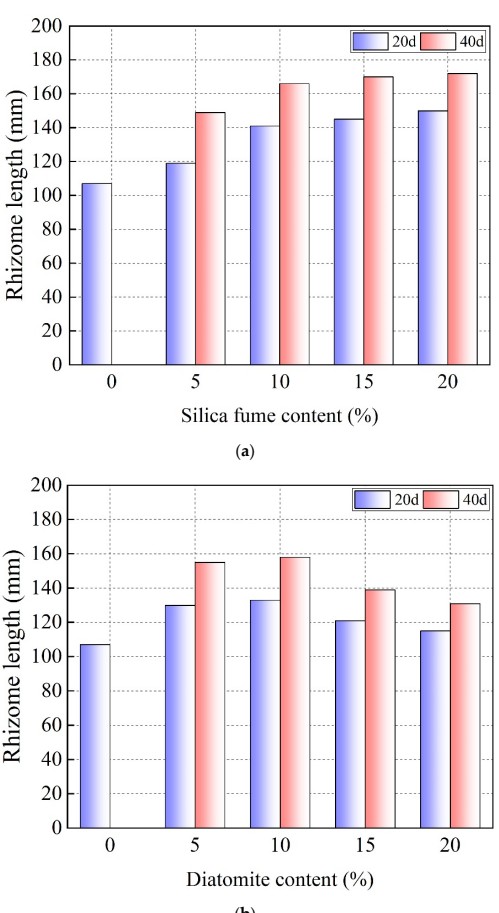

**Figure 17.** Growth status of plants with different contents of SF and DA: (**a**) SF; (**b**) DA.

Figure 17b shows that when the DA content exceeds 5%, the plant growth rate slows down. When the content exceeds 10%, the improvement effect of DA on the planting performance of PC decreases. Although Figures 10 and 13 showed that too much DA will not have a great impact on the pore characteristics of PC, some pores are too narrow, resulting in the high alkalinity of the local soil and affecting the plant's early growth rate. Too narrow local pores will also make some roots of plants unable to absorb enough nutrients, resulting in a significantly hindered plant growth rate in their later stages of growth. From the growth status of the final plant, when the SF content is greater than 5% and the DA content is maintained at 5% or 10%, PC was observed to exhibit good planting performance. Compared to the PC without SF and DA, DA and SF increased the planting performance of PC by about 40.2% and 24.3%, respectively.

### 3.7. Slope Stability Finite Element (FE) Analysis

PC can also be used for slope protection. Whilst strengthening the slope, it can also simultaneously aid in the conservation of the environment. When PC is used for slope protection, lattice reinforcement is generally carried out first prior to introducing the PC. Using PC with a A/P ratio of 0.2 and SF content of 10% as an example, the slope protection model was established (see Figures 18 and 19), and a finite element analysis was carried out.

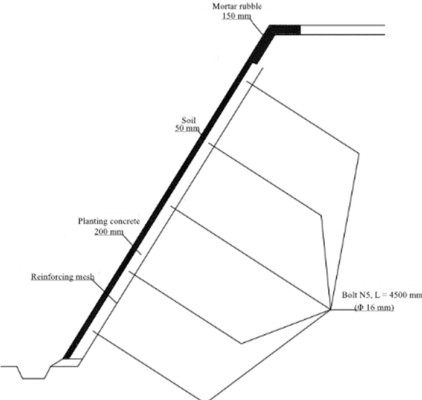

**Figure 18.** Side slope section.

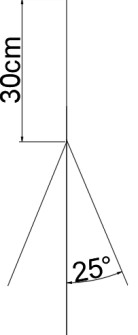

**Figure 19.** Two dimensional model of the plant roots.

In this study, ABAQUS was used to analyze the slope protection performance of the PC [46–48]. The results of these FE analyses are shown in Figures 20 and 21. Compared with the stress diagram before and after slope protection (Figure 20), the maximum stress of the slope before protection is 178.8 kPa at the bottom of the slope, and the maximum stress of the slope surface is 90.8 kPa at the foot of the slope. After PC protection, the maximum stress area of the slope is transferred from the slope bottom to the PC protection area of the slope surface. The peak stress at the slope corner is 448.6 kPa, and gradually decreases from the slope toe to the top.

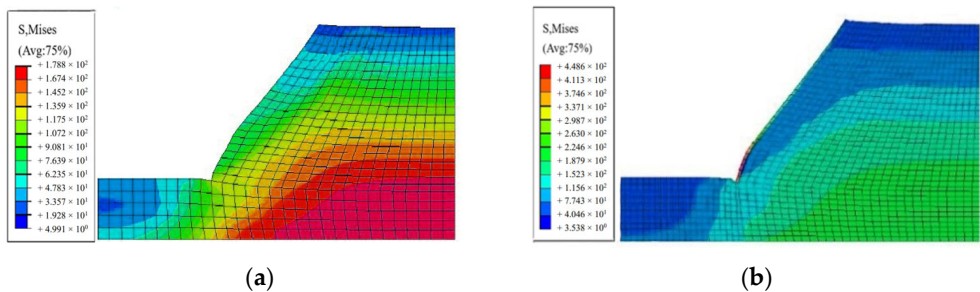

**Figure 20.** Stress profile of the slope from FE analysis (**a**) before protection inclusion; (**b**) after protection inclusion.

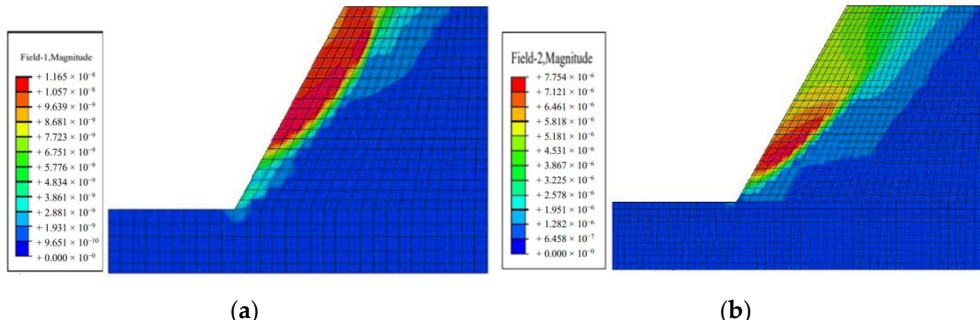

**Figure 21.** Slope sliding surface from FE analysis (**a**) before protection inclusion; (**b**) after protection inclusion.

It can be seen from the change of slope sliding surface before and after PC protection in Figure 21 that, before protection, the sliding surface is mainly concentrated on the upper slope; after protection, the sliding surface of the slope body is significantly reduced, mainly distributed at the lower slope surface and slope toe, and the sliding angle is significantly reduced, significantly reducing the risk of slope sliding. Therefore, the application of PC in slope protection not only protects the slope but can also conserve and beautify the environment [49].

## 4. Conclusions and Recommendations

The work presented in this paper studied the effects of SF and DA on the properties of red mud geopolymer PC under different A/P ratios. From the laboratory tests and FE modeling that were conducted, the following conclusions were drawn:

- When the content of SF and DA is 10% and 5%, respectively, the PC compressive strength quantittatively reached its best performance and increased with an increase in the A/P ratio;
- The pore characteristics of PC were found to be mainly related to the fluidity of geopolymer slurry. For good PC pore characteristics, the fluidity of the mortar should be 112~128 mm. When the SF and DA contents are less than 10% and 5%, respectively, their morphological effect was observed to significantly improve the fluidity of the slurry. However, if the content continues to increase above these levels, the high water absorption capacity of SF and DA, along with the high plasticity of DA, will significantly retard the flow of the slurry, i.e., reduce slurry fluidity;
- A/P was found to be the main factor affecting the soil alkalinity. When the A/P ratio is not higher than 0.2, adding SF and DA can lower the soil pH value to less than 9. In the early stages of plant growth, soil alkalinity was observed to be the main influencing factor. The later growth of the plants was oberseved to be mostly affected by the PC pore characteristics;
- Considering the overall performance of PC, the optimum A/P ratio was found to be 0.2 at 10% SF and 5% DA;

- Through finite element analysis of the PC slope stability, it was found that geopolymer modifed PC can effectively improve the slope stability, whilst also contributing to environmental protection and beautification. In general, the study has demonstrated that it is feasible to prepare PC with red mud geopolymer instead of OPC and concurrently use it in slope protection applications.

Overall, the findings reported herein have demonstrated that it is feasible to prepare PC with geopolymer instead of OPC and provides a new direction for the potential treatment/applications of red mud. However, this paper did not evaluate the durability of PC, nor did it study the impact of PC alkalinity diffusion on the surrounding environment. This needs to be further explored in future research, including the application of PC in slope protection in practical projects, to verify the final results of PC finite element slope protection analysis.

**Author Contributions:** Writing—original draft, W.C.; Writing—review and editing, W.C. and J.L. All authors have read and agreed to the published version of the manuscript.

**Funding:** This research was funded by the Postgraduate Scientific Research Innovation Project of Hunan Province (CX20210755) and the Natural Science Foundation of Changsha (kq2007025).

**Institutional Review Board Statement:** Not applicable.

**Informed Consent Statement:** Not applicable.

**Data Availability Statement:** Not applicable.

**Conflicts of Interest:** The authors declare no conflict of interest.

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
