# Peer review of "Effects of Different Silicon Sources on the Properties of Geopolymer Planting Concrete Mixed with Red Mud"

_sustainability, doi:10.3390/su15054427_

Round 1
Reviewer 1 Report
This manuscript is presented by research at a fairly high level and has a certain novelty and relevance. A very interesting research topic, which is devoted to the Effects of Different Silicon Sources on the Properties of Geopolymer Planting Concrete Mixed with Red Mud. At the same time, there are a number of comments on the submitted article:
1) In the manuscript in the introduction, standard Portland cement is used and given, however, its brand, manufacturer, city and country are not specified. It is also not specified, its basicity, on which the processes of concrete hardening in 3, 7, and 28 days depend, with different bases of Portland cement, the setting time on its basis will be different, as will the entire subsequent hardening process. In this regard, it is necessary to consider similar works of foreign scientists in the manuscript, in particular:
1.1) Kolesnikova O., Vasilyeva N., et all. Optimization of raw mix using technogenic waste to produce cement clinker. MIAB. Mining Inf. Anal. Bull. 2022;(10-1):103—115. [In Russ]. https://doi.org/10.25018/0236_1493_2022_101_0_103 .
1.2) Kolesnikova, O.; Syrlybekkyzy, S.; Fediuk, R.; Yerzhanov, A.; et all. Thermodynamic Simulation of Environmental and Population Protection by Utilization of Technogenic Tailings of Enrichment. Materials 2022, 15, 6980. https://doi.org/10.3390/ma15196980
1.3) Amran, M.; Fediuk, R.; Murali, G.; Vatin, N.; Karelina, M.; Ozbakkaloglu, T.; Krishna, R.S.; Kumar, S.A.; Kumar, D.S.; Mishar, J. Rice Husk Ash‐Based Concrete Composites: A Critical Review of Their Properties and Applications. Crystals 2021, 11, 168. https://doi.org/ 10.3390/cryst11020168
2) The origin of red mud bauxite, blast furnace slag and fly ash, which are used in experiments, is also not indicated. It is necessary to specify the organization, city and country of origin of these wastes. And their accumulated amount, as well as to note about their harm to the environment and public health.
3) In the third paragraph of the introduction, you need to change the font of the text.
4) The authors provide phase analyses of SF and DA, but there is no information and analyses on red mud and BFS. It is necessary to add this information and indicate the phases available in blast furnace slag and red mud, as this is of scientific interest to readers and researchers.
5) The signatures of axes and captions in Figures 1, 3-20 have a different font and font size, which does not correspond to the style of the journal design. These inaccuracies must be eliminated in accordance with the requirements of the design of this journal.
6) Literary sources are not designed in accordance with the requirements of the journal, it is necessary to design in accordance with the requirements. Needs to be fixed.
Author Response
We feel great thanks for your professional review work on our article. As you are concerned, there are several problems that need to be addressed. According to your nice suggestions, we have made extensive corrections to our previous draft. Please refer to the attachment for details.

Reviewer 2 Report
1. The novelty of this work is unclear. In the last paragraph of introduction section be added more explanation.
2. The manuscript needs English polishing.
3. A flowchart for showing the fabrication process and performed tests be added.
4. Cracks in Fig. 4 be marked.
5. How many samples have been considered for repeating each test? Be added into the manuscript.
Author Response

(The authors gave the same response as above.)

Reviewer 3 Report
The disposal of red mud is very topic issue. Solutions for the efficient use of red mud by partially replacing natural materials are still being sought. A very promising and innovative direction is shown in this study. Partial replacement of Portland cement, as well as its properties regulation with by-products such as red mud and fly ash are very promising. Especially when the environmental impact gets more and more attention. Plants are increasingly being used in engineering for environmental impact assessment, which is also done in this study. However, plant growth or non-growth is only a partial indicator, as the plant also absorbs substances harmful to animals. Therefore, it would be desirable to evaluate this factor in the future. The temperature at which the samples are held is not specified, whether it is 20±2 degrees Celsius? The origin and mineralogical composition of the aggregates also play an important role, which can affect the properties of the curried material. The application of red mud in road materials/road pavements is studied very extensively, as well as environmental assessments have been carried out using various approaches, so it is recommend including this factor as well, for example, referring to the DOI: 10.1080/14680629.2021.1900899
Author Response

(The authors gave the same response as above.)

Reviewer 4 Report
In general, the paper is well written. However, the following comments should be addressed before the paper can be accepted.
1. In Introduction, Paragraph 6, before “In this study, two high silicon sources, ….”, please add one sentence to state that silica fume and diatomite are the materials that contain high silicon by referring the following papers:
a) Alkhaly, Y.R.; Abdullah; Husaini; Hasan M. Characteristics of reactive powder concrete comprising synthesized rice husk ash and quartzite powder. J Cleaner Prod. 2022, 375, 134154. https://doi.org/10.1016/j.jclepro.2022.134154
b) Hasan, M.; Saidi, T.; Muyasir, A.; Alkhaly, Y.R.; Muslimsyah, M. Characteristics of calcined diatomaceous earth from Aceh Besar District–Indonesia as cementitious binder. IOP Conf. Ser. Mater. Sci, Eng. 2020, 933, 012008. https://doi.org/10.1088/1757-899X/933/1/012008.
c) Saidi, T.; Hasan, M. The effect of partial replacement of cement with diatomaceous earth (DE) on the compressive strength and absorption of mortar. J. King Saud Univ. Eng. Sci. 2022, 34, 250–259. https://doi.org/10.1016/j.jksues.2020.10.003.
d) Hasan, M.; Saidi, T.; Husaini, H. Properties and microstructure of composite cement paste with diatomaceous earth powder (DEP) from Aceh Besar district–Indonesia. Asia-Pacific J. Sci Technol. 2022, 27, APST-27-01-03. https://doi.org/10.14456/apst.2022.3.
2. Section 2.2. “…… and water were poured with a water/cement ratio of 0.3. Was the cement used in those mixtures? If no, the term water/cement should be replaced with another term.
3. Section 2.3.2. “The diameter” should be replaced with “the flow diameter”.
4. Section 2.3.6. Please add the figure of FE model analysis.
5. Please cite Fig. 5 in the text and add the discussion on the results of Fig. 5 (a-f), not only Fig. 5(d).
6. Fig. 7 caption. Which red line do you mean since there are 2 red lines in the figures. Is it above 10 MPa. Please revise for clarity.
Author Response

(The authors gave the same response as above.)

Reviewer 5 Report
The article presents the effect of the content of alkaline solids and powder dosage as well as additives with a high content of silica (silica fumes and diatomite) on the microstructure and fluidity of the geopolymer, as well as the compressive strength, pore characteristics and alkalinity of concrete. It is incomprehensible why the cylindrical samples were not tested to determine the modulus of elasticity of concrete. The confirmation of the slope stability simulation requires clarification.
In its current form, the article is not suitable for publication in Sustainability.
General remarks
1 Explain what effect PC has on the subsoil material.
2. The authors do not provide the characteristic values of the compressive strength taking into account the standard deviation. This should be included in the article.
3. The applied FEM model of plant roots raises doubts. It should be clearly stated whether the FEM simulation results have been tested.
4. How was the stability tested? If so, please provide the results.
The article is not well written and needs improvement.
Specific remarks
5.4 point 2.3.3: Explain how the samples were tested. Why were cubes with a side of 150 mm not chosen, whose compressive strength corresponds to real conditions due to the scale effect?
6. 5 p. Equation 2: Mismatch between the units ms (g) and Cd (%)?
7. 5 p. The relationship between PC and humus soil should be clarified here.
8. 5 p. Please explain the purpose of the measurement of the plant rhizome length?
9. 5 p. Authors in Tab. 3 give soil parameters, what about PC parameters?
10.5 point 3.1.1 Expand the abbreviation XRD.
11. Fig.17 : Illegible drawing; should be improved.
12. Conclusions should be bulleted.
I recommend reviewing the manuscript with comments to make it suitable for publication in Sustainability.
Author Response

(The authors gave the same response as above.)

Reviewer 6 Report
The manuscript, entitled "Effects of Different Silicon Sources on the Properties of Geopolymer Planting Concrete Mixed with Red Mud," presents an interesting experimental study conducted on the effect of different activator/aluminosilicate ratio on the main properties of red mud based geopolymers. However, the paper needs major revisions before it is processed further. Some comments follow: Title: The title is too long and unclear. Please consider replacing the title with a clear formula that reflects the content of the manuscript. Abstract: Please briefly describe the novelty of the study in the abstract. Introduction section As the author’s state: "A geopolymer-based PC with excellent performance can be produced from red mud." This is well known in the literature; therefore, what is the novelty of the study? Please highlight what makes this research new in the literature. Experimental matrix plan section Table 1: Two types of iron oxides have been detected in these types of materials; therefore, please replace Fe2O3 with FexOy in the XRF analysis or provide the scientific proof to support your results (XRD that show the presence or absence of magnetite (Fe2O3) or hematite (Fe3O4)). Morphology of SF and DA: As presented, there are no morphological characteristics that can be analyzed or observed from these images. Please provide SEM analysis or, at least, optical microscopy. The raw materials are insufficiently characterized to assure the experiments' repeatability. Please provide LOI, specific surface area, density, grading for each material involved in the experiment. Mix proportions of PC – There is no rationale in chosing the amount of each constituent. Please briefly describe why the authors used these mixtures instead of others. The XRD isn’t used to characterize the microstructure of the materials, but their mineralogical characteristics. Results, analysis, and discussion XRD analysis: why authors considered some peaks instead of others Please evaluate all peaks that appear on the XRD spectra for each sample. The XRD spectra of the analysed sample show some clear peaks around 22,24,28,34,36,47, 2θ º that weren’t considered. Also, there are some clear differences between the spectra of the samples that are definitely worth discussing. SEM analysis: the indications and labels in the images are not relevant. Please indicate the areas of interest in accordance with the discussions. Which are the differences that have been observed during the microstructural analysis? Cracks can be seen in all samples, and the areas indicated as aggregates seem misplaced. Please confirm the components indication by EDS. Fluidity test and compressive strength—How many samples have been tested from each batch? Please provide the measurement values with a deviation bar. Figure 8: Please introduce a scale bar on each figure. Discussion section. The discussion section is missing. In the discussion section, a clear correspondence and comparison between the results of this study and those in the literature should be provided. Please improve them, compare the obtained results with those from the literature, and make qualitative and quantitative evaluations. Future directions and limitations: Please provide some future directions and limitations of the study.
Round 2
Reviewer 1 Report
All comments were answered by the authors and changes were made to the current paper.
Author Response
Reviewer 1
General Comments:
All comments were answered by the authors and changes were made to the current paper.
Response: We appreciated your warm work earnestly and recognition. Thank you!
Reviewer 4 Report
The authors have revised the manuscript based on my previous comments. Only revising comment 3 is still not satisfactory to me. What I mean with comment 3 is to revise the sentence into:
"The fluidity of the geopolymer mortar with different SF and DA contents was determined by measuring the flow diameter of the geopolymer using a cone (60 mm high, 36 mm top diameter and 60 mm bottom diameter)."
Author Response
Reviewer 4
General Comments:
The authors have revised the manuscript based on my previous comments. Only revising comment 3 is still not satisfactory to me. What I mean with comment 3 is to revise the sentence into:
"The fluidity of the geopolymer mortar with different SF and DA contents was determined by measuring the flow diameter of the geopolymer using a cone (60 mm high, 36 mm top diameter and 60 mm bottom diameter)."
Response: We have revised it according to your comments. We sincerely thank you for your enthusiastic work and recognition. Thanks!
Reviewer 6 Report
Dear authors,
You have done a great job in revising the paper according to our sugestions. I have only one suggestion.
Figure 9: Please introduce a scale bar on each figure.
Author Response
Thank you for your suggestions. Please refer to the attachment for details.
